# Impact evaluation of a cash-plus programme for children with disabilities in the Xiengkhouang Province in Lao PDR: study protocol for a non-randomised controlled trial

Lena Morgon Banks [1], Bounhome Soukkhaphone,[2] Nathaniel Scherer [1], Latsamy Siengsounthone,[2] Mark T Carew,[1] Tom Shakespeare,[3] Shanquan Chen,[1] Calum Davey,[1] Divya Goyal,[4] Anja Zinke-Allmang,[5] Hannah Kuper [6], Ketmany Chanthakoumane[2]

HK and KC contributed equally.

LMB and BS are joint first authors.

For numbered affiliations see end of article.

**Correspondence to**
Dr Lena Morgon Banks; morgon.banks@lshtm.ac.uk

## ABSTRACT

**Introduction** More than 170 countries have implemented disability-targeted social protection programmes, although few have been rigorously evaluated. Consequently, a non-randomised controlled trial is being conducted of a pilot 'cash-plus' programme implemented by UNICEF Laos and the Laos government for children with disabilities in the Xiengkhouang Province in Laos. The intervention combines a regular cash transfer with provision of assistive devices and access for caregivers to a family support programme.

**Methods and analysis** The non-randomised controlled trial will involve 350 children with disabilities across 3 districts identified by programme implementers as eligible for the programme (intervention arm). Implementers have also identified approximately 180 children with disabilities in neighbouring districts, who would otherwise meet eligibility criteria but do not live in the project areas (control arm). The trial will assess the impact of the programme on child well-being (primary outcome), as well as household poverty, caregiver quality of life and time use (secondary outcomes). Baseline data are being collected May–October 2023, with endline 24 months later. Analysis will be intention to treat. A complementary process evaluation will explore the implementation, acceptability of the programme, challenges and enablers to its delivery and mechanisms of impact.

**Ethics and dissemination** The study has received ethical approval from the London School of Hygiene and Tropical Medicine and the National Ethics Committee for Health Research in Laos. Informed consent and assent will be taken by trained data collectors. Data will be collected and stored on a secure, encrypted server and its use will follow a detailed data management plan. Findings will be disseminated in academic journals and in short briefs for policy and programmatic actors, and in online and in-person events.

**Trial registration number** ISRCTN80603476.

## STRENGTHS AND LIMITATIONS OF THIS STUDY

⇒ The study design is robust, combining a non-randomised controlled trial and a process evaluation, to assess if and how a novel cash-plus programme impacts children with disabilities and their households' well-being.
⇒ Trial design involved feedback from multiple stakeholders to ensure outcomes being investigated are in line with the programme's intended aims.
⇒ Still, individuals could not be randomised and so there may be unobservable differences between intervention and control arms.
⇒ Similarly, the sample size likely will not have sufficient power to explore meaningful differences among children with disabilities (eg, if impacts differ by the child's gender, disability type, etc).

## INTRODUCTION

Social protection is an important tool for reducing poverty and protecting people from risks that affect well-being across the lifecycle.[1 2] People with disabilities, who comprise 16% of the global population, are a core target group for social protection.[3 4] People with disabilities have a right to social protection under the United Nations Convention on the Rights of Persons with Disabilities, which defines disability as including '… those who have long-term physical, mental, intellectual or sensory impairments which in interaction with various barriers may hinder their full and effective participation in society on an equal basis with others'.[5]

Furthermore, people with disabilities may have a higher need for social protection. For example, they are more likely to be living in poverty compared with people without disabilities.[6 7] People with disabilities are

often excluded from opportunities for decent work, due to discrimination or earlier barriers to receiving quality education and training, while caregivers often reduce time on income-generating activities for additional care-taking responsibilities.[8] They also frequently face 'extra costs' for disability-related goods and services (eg, rehabilitation, assistive devices, personal assistance).[9] As such, people with disabilities and their households often require a higher level of income than people without disabilities to meet both basic needs and disability-related costs.[10]

Many countries have recognised the need for and right to social protection among people with disabilities, with over 170 countries implementing disability-targeted social assistance programmes.[4] The coverage and content of these programmes can vary widely: for example, only 9% of people with severe disabilities receive a disability benefit in low-income countries compared with 86% of people with severe disabilities in high-income countries.[11] However, there is a lack of evidence on the impact of social protection among people with disabilities.[12 13] A systematic review on social protection and disability in low-income and middle-income countries published in 2017 retrieved few rigorous studies on the impact of cash transfers and other programmes among people with disabilities, with available evidence limited to cross-sectional or qualitative studies of self-reported impacts with no impact evaluations.[12] Since then, impact evaluations of cash transfers among people with disabilities have been conducted for disability-targeted programmes in the Maldives and China,[14–16] as well as mainstream programmes in Malawi and Lesotho.[17] Among people with disabilities of all ages, impacts of cash transfers were modest and mostly concentrated on improvements in food security, health status and increased health-seeking behaviour.[14 17] Further evidence is needed on the impact of different programmes from different settings, and among children with disabilities and their households in particular.

Additionally, there is an increasing recognition that cash transfers are an important but insufficient tool to push people out of poverty and improve their well-being.[18] Cash transfers alone may be particularly insufficient for people with disabilities and their households due to the poor availability, quality, accessibility and affordability of needed goods and services.[19] For example, assistive devices—such as wheelchairs, prosthetics, hearing aids, walking canes—are essential for improving functioning and participation of people with disabilities; however, in many settings, people with disabilities lack access to these devices because their supply is very limited, they lack information about their utility and where to access them and available devices are of poor quality.[20] Additionally, children with disabilities and their households may face other barriers to improving their well-being, such as discrimination and negative attitudes on disability as well as lack of inclusive health, education and other services.[8] Caregivers may lack knowledge about disability and how to best support their child with a disability, which can lead to increased caregiver stress, less effective caregiving and ultimately poorer childoutcomes.[21]

As such, there is an increasing focus on developing integrated social protection or 'cash-plus' programmes. Cash-plus programmes provide recipients with both a cash transfer as well as complementary programmes, such as health insurance and linkages to key services.[22] These programmes show particular promise for children and adults with disabilities, who often face multiple barriers in improving their livelihoods and well-being.[8] People with disabilities report spending cash transfers primarily on healthcare,[23 24] so cash-plus programmes with entitlements linked to health are likely to be relevant. In China, receipt of a disability-targeted cash transfer was linked to an increased likelihood of using rehabilitation and medical services among both children and adults with disabilities.[15 16]

Consequently, this study will conduct an impact evaluation of a cash-plus programme targeted to children with disabilities in Laos. According to the 2015 Population and Housing Census, 2.8% of the population over 5 years of age in Laos is living with a disability, with prevalence higher in remote areas.[25] The Lao People's Democratic Republic (PDR) Constitution and the Law on Persons with Disabilities stipulates equal protections of the rights of people with disabilities, and the latter calls for the creation of a welfare fund for people with disabilities.[26] Overall, Laos' social protection system is nascent, but growing. For example, health insurance coverage expanded rapidly from 11% in 2008 to 93% currently.[27] It covers a range of preventative, promotive, curative and some rehabilitation services. However, it is acknowledged that insufficient state financing of the health system has led to inadequate services and high out-of-pocket payments.[27] Laos does not currently have comprehensive social protection programmes for people with disabilities, although expanding services and benefits to this group is listed as priority for action in the most recent National Social Protection Strategy.[26]

This impact evaluation will assess a pilot of one such programme. The pilot programme combines a regular cash transfer with an assessment for and provision of rehabilitation and assistive devices. Caregivers are also invited to join a family support programme, which focuses on improving their livelihood skills and on learning more about disability and how to best support their child with a disability. The cash-plus programme will be run by UNICEF Laos in partnership with the Laos government in three districts of the Xiengkhouang Province in Lao PDR. An accompanying process evaluation will also explore the implementation process, acceptability of the programme, challenges and enablers to its delivery and mechanisms of impact. This information will be critical for the Lao PDR government, given their prioritisation of disability within strategies for scaling-up the national social protection system.[26] It will also have implications for other contexts, given the lack of evaluations on disability-targeted cash-plus programmes.

## Objectives

The overall aim of this research is to assess the impact of a cash-plus programme among children with disabilities and their households in Laos.

Specific objectives include:

► To evaluate the impact of a cash-plus programme on children's well-being.
► To assess the impact of a cash-plus programme on caregivers' well-being.
► To measure the impact of a cash-plus programme on household consumption expenditures.
► To estimate the impact of a cash-plus programme on unmet needs for disability-related goods and services.
► To explore the perceived impact, and acceptability of the cash-plus programme among children with disabilities and their caregivers.
► To examine the implementation process, challenges and enablers to the delivery and mechanisms of impact of the cash-plus programme from the perspective of children/caregivers and implementers.

## METHODS

### Study design and settings

The trial will be a superiority, non-randomised controlled trial with two parallel groups. Three districts (Pek, Khoun, Phoukoud) in the Xiengkhouang Province were selected by UNICEF Laos and the Laos government to implement the cash-plus programme. Two other neighbouring districts were selected as control areas (Nonghed, Kham) after consultation between the research team (London School of Hygiene and Tropical Medicine (LSHTM), Laos Tropical and Public Health Institute (Laos TPHI)) with UNICEF Laos as these areas have similar sociodemographic characteristics to the intervention districts.

Three districts (Pek, Khoun, Phoukoud) in the Xiengkhouang Province were selected by UNICEF Laos and the Laos government to implement the cash-plus programme.

Two other neighbouring districts were selected as control areas (Nonghed, Kham) after consultation between the research team (LSHTM and Laos TPHI) with UNICEF Laos as these areas have similar sociodemographic characteristics to the intervention districts.

A non-randomised controlled trial design was used as it was not possible to randomise children to receiving or not receiving the cash-plus programme, as the programme implementers had logistical constraints that meant they could only operate the programme in the selected areas at this time.

Baseline data are being conducted between May and October 2023. Collection of data from the intervention group was completed before they began receiving any part of the programme. Allocation into the programme was known by implementers and the research team a priori, although participants were not aware that they would be offered enrolment into the programme before baseline data collection. Delivery of the intervention began in September 2023, and parts (ie, cash transfer) will be continued indefinitely depending on funding. Endline data collection will take place 24 months from baseline. The process evaluation will be conducted in 2024 after the delivery of the family support programme and assistive device components.

### Intervention description

The intervention includes a cash transfer along with an assessment for assistive devices (eg, wheelchair, prosthesis, hearing aid, walking cane). Children found to have an unmet need for assistive devices will be provided with devices free of charge as part of the programme. Caregivers will also be invited to join a family support programme, which aims to improve their understanding of their child's disability, how to provide support to their child and financial literacy/livelihoods. Figure 1 presents

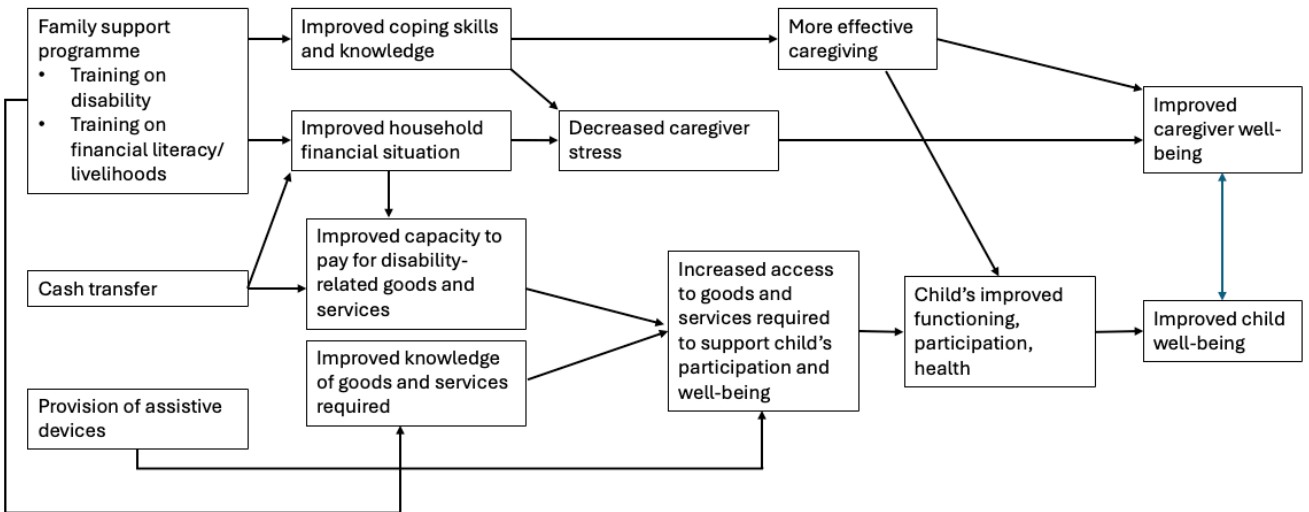

**Figure 1**  Theory of change for cash-plus programme.

a theory of change for how the intervention components are expected to improve child and caregiver well-being.

The programme is being delivered by UNICEF Laos and the Laos government. They currently have funding to pilot the programme for approximately 450 children with disabilities in the Pek, Phoukoud and Khoun districts of Xiengkhouang Province. The intervention is expected to continue to be offered to participants for the duration of the study period. This timeline may change if implementers face budgetary, political or other constraints in continuing to operate all or part of the programme. Funding is guaranteed until June 2024, and will be reassessed for continuation before then. Participants are free to disenroll from the programme at any time. Participants in either arm are free to take up any other programme that is offered to them during the study period.

### Eligibility criteria and recruitment

Data were collected from children with disabilities and their caregivers. To be eligible for this study, children had to be living in one of the study districts at the time of baseline data collection, be 18 years of age or younger and be identified by UNICEF Laos/Laos government as having a disability. Disability determination and identification for both the intervention and control groups was conducted by UNICEF Laos. An initial sample—which strove to identify all children with disabilities in the study areas—involved a combination of past survey data shared by the government of Laos and identification conducted by community leaders; these children were then verified by UNICEF to ensure they had a disability and were of the target age. Further details on the identification process will be explored as part of the process evaluation.

For the process evaluation, approximately 15 caregivers of children with disabilities from the intervention group will be interviewed. They will be selected to maximise heterogeneity by the child's gender, disability type, age and location. Additionally, about 12 programme implementers will be interviewed. They will be selected to reflect diversity of roles in the design and delivery of the programme.

Participants (primarily caregivers of children with disabilities) were invited to participate in this study by members of the research team through either in-person or phone communication. Participants were provided with, or read out, an information sheet detailing the study protocol and risks and benefits of participating. Participants were provided with compensation to offset their time and other expenses (eg, phone data) of taking part in the research.

### Outcomes

#### Primary outcome measures

The primary outcome measure is child well-being, which will be measured across multiple dimensions. Multidimensional Overlapping Deprivation Analysis (MODA) have often been used for this purpose.[28] These indicators are also in line with Sustainable Development Goal (SDG) Indicator 1.2.2, which focuses on reducing poverty 'in all its forms'.[29]

MODAs are indexes that collate several variables on deprivation. They can be tailored for the context and population group of interest.[28] The index for this study was created to be relevant to children with disabilities in Laos across the years of childhood (0–17 years) and covers domains that could be feasibly affected by receipt of the cash-plus programme (online supplemental appendix 1). It included commonly used measures, such as ones used in the Multiple Indicator Cluster Surveys that have been administered in over 100 countries including Laos. It was defined through consultations with UNICEF Laos, Laos government and representatives of Organisations of Persons with Disabilities in Laos. Each indicator will be given equal weight such that the total of the index falls between 0 and 1 (0 being child experiences no deprivation on any indicator, 1 being child experiences deprivation across all indicators).

#### Secondary outcome measures

We will explore household consumption expenditures across food and non-food items. We will compare differences in overall consumption expenditures and by category (food, non-food, health, education). If possible, poverty will be measured by comparing consumption expenditures against the national poverty line. This measure is in line with SDG Indicator 1.2.1.[29]

Caregiver well-being will be measured using the Pediatric Quality of Life (PedsQL) Family Impact Module, which looks at caregiver and family strain linked to having a child with a health condition.[30] It has been used in different settings for measuring well-being of caregivers of children with disabilities.[31 32] A score will be computed according to the tool's scoring guidance.[33] Caregiver time use across different activities will be assessed using a modified version of the 'stylised activity list' developed for the Living Standard Measurement Survey.[34]

Finally, we will assess unmet need for different disability-related goods and services by creating a module on the caregiver-reported need for and use of different goods and services required to support their child's participation (eg, healthcare, assistive devices, inclusive education supports, personal assistance). For each item, caregivers will be asked if they think their child needs the good/service, if they have gone for the service and their satisfaction with that service.

#### Process evaluation

The process evaluation will follow the Medical Research Council (MRC) framework.[35] It will examine the structures and resources used to deliver the intervention and what was delivered, for example, in terms of whether the intervention reached all whom it was designed to reach.

It will also explore mechanisms of impact, including participants' reflections on the strengths and challenges of different components and their perceptions of the impact of the programme (or lack thereof) on their lives,

children and personal well-being. The latter will complement the quantitative measures of impact, as concepts such as well-being are complex and challenging to measure. Finally, the process evaluation will explore the role of context, for example, how underlying socioeconomic factors, social attitudes to disability and broader disability inclusion in the study areas may have influenced the effectiveness of the programme.

## Sample size

The primary outcome measure is child well-being, measured through a MODA index. Disability-sensitive MODAs are lacking, but using data from a previous study in Vietnam as parameters,[15] we assume an average score of 0.4 (score out of 1) and an SD of 0.12. With a sample size of 330 in the intervention arm and 180 in control arm, we can detect a minimum change in score of 0.035 (1/3 of an SD) with power of 80%, type 1 error of 5% and 20% loss to follow-up.

The process evaluation will include in-depth interviews with about 15 caregivers of children with disabilities and 12 implementers.

## Data collection and management

Baseline data will be collected between May and October 2023. Endline data will be collected 24 months later. All outcomes will be collected through a mobile survey created through Open Data Kit (ODK) to promote data quality. The survey tool has built-in skips and generates error messages for missing data and for non-logical answers (eg, answers to daily time use questions exceeding 24 hours). The survey was developed by researchers at LSTHM and Laos TPHI, using validated tools (eg, PedsQL, time use adapted from Living Standards Measurements Surveys) where possible. It was translated from English into Lao and Hmong. The translated versions were reviewed by multiple native speakers and pilot tested prior to implementation.

The survey will be administered by trained data collectors, who underwent a weeklong training in the study protocols, practice using the questionnaire and on how to promote quality data collection. They are predominantly public health students at the Lao TPHI and speak Lao and/or Hmong.

The primary caregiver of each child with disabilities will be the main respondent, although children with disabilities aged 5–17 years may answer certain modules (eg, child's social participation, experiences in school). Other household members besides the primary caregiver may also contribute to sections on household consumption expenditures if they are the most knowledgeable household member. Multiple means of contacting participants were collected during baseline to decrease loss to follow-up by endline.

The process evaluation will be conducted in 2024 after the completion of the family support programme and delivery of assistive devices. In-depth interviews with caregivers and implementers will be conducted by trained interviewers in Lao, Hmong or English depending on the preference of the participant. They will use semi-structured interview guides. For implementers, these guides will be adapted to reflect the expertise and role of the respondent.

## Data management

Data for the trial will be collected and stored on a secure and encrypted ODK server, hosted by LSHTM. In-depth interviews will be recorded, translated and transcribed. All data will be cleaned and anonymised by an experienced data manager at Lao TPHI. Hard copies of data (eg, written consent forms) will be kept in locked, secure cabinets at Laos TPHI. Electronic data (eg, survey responses, interview recordings) will be kept on a password-protected computer and on secure servers. A master file linking participants' names with ID numbers, signed consent forms and contact/locator information will be stored in as an encrypted file on a password-protected computer that is separate from the data files. Non-anonymised data will be destroyed 2 years after study completion.

Twenty-four months after the end of the project, the data may be made available on LSHTM's Data Compass if it can be done so in a way that both allows for replication and does not lead to potential identification of participants. The possible identification of participants is a concern given that a relatively small number of children are receiving the cash-plus programme: combining basic details such as age, gender, disability type and area of residence may be sufficient to identify a participant. The Data Protection Office at LSHTM will be consulted about the risk of identification.

If access to the data is possible, it will be made available on request, provided the requester has a legitimate purpose for using the data. Requests will be managed by LSHTM's Data Compass team, which is independent to the research team.

## Patient and public involvement

Preliminary outcome indicators were checked with the Lao Disabled People's Association, an Organisation of Persons with Disabilities, to confirm relevance for people with disabilities in Laos. Findings from the study will be disseminated back to participants (eg, through community events).

## Analysis

A detailed analysis plan will be published before the endline survey. The analysis will estimate intention-to-treat (ITT) effects.

We will assess the quality of the balance between control and intervention groups by describing the primary and secondary outcomes and sociodemographic variables at baseline. If there is evidence of imbalance, based on subjective interpretation of the magnitude of the difference, then we will plan a priori to include such variables in the main analysis of the primary and secondary outcomes.

We will estimate the effect of the intervention by comparing the proportions (eg, proportion of children deprived on each domain of the MODA) and the means/medians (eg, total MODA score, household consumption expenditures) between the arms of the trial. We will report the unadjusted estimated effects as risk ratios for binary outcomes and difference in the means for continuous outcomes. In the final analysis, to increase the precision of the estimates and reduce the risk of bias from imbalances at baseline, we will use regression to adjust for the baseline levels of the outcome, stratification variables and variables considered to be imbalanced at baseline. For binary outcomes, we will model the risk ratio with a modified Poisson regression model.[36] For continuous outcomes we will use linear regression. We will disaggregate findings by gender and other characteristics if the final sample sizes are sufficiently powered.

For the process evaluation, thematic analysis will be used (Braun and Clarke, 2012).[37] Transcripts of in-depth interviews will be analysed both deductively and inductively in NVivo. Deductive analysis will use a coding framework based on the core areas of the MRC process evaluation framework, with inductive coding being used to capture additional themes identified by analysts in the data. Analysis will consider differences by subgroups, such as by the gender or disability type of the child.

### Methods in analysis to handle protocol non-adherence and any statistical methods to handle missing data

Analysis will be ITT. The programme implementers do not have plans to expand the delivery of the cash-plus programme to the control areas or to offer the programme to people newly moving into the intervention areas during the study period, so spill-over is unlikely. In the unlikely event that the programme is expanded to control areas, we may reduce the period of follow-up.

Missing data are expected to be minimal given the quality checks in the questionnaire (ie, built-in skip patterns, error messages for unentered data). Some remaining missing data can be imputed. For example, for consumption expenditures data, missing items or items where participants could not give an accurate estimate of value will be imputed based on the median reported cost of that item from other households in the area.

### ETHICS AND DISSEMINATION

Ethical approval has been received from the institutional review boards at the London School of Hygiene and Tropical Medicine, UK (reference: 28234) and from the National Ethics Committee for Health Research in Laos (submission ID: 2023.03). Any amendments will be approved by these bodies, and uploaded into the trial registry (https://www.isrctn.com/ISRCTN80603476).

Informed consent and assent will be sought before the start of all interviews (ie, survey, qualitative in-depth interviews) by trained data collectors. Consent will be sought from all adults (aged 18+ years) providing data.

Caregivers who are married and between 14 and 17 years will also provide their own consent per Laos law and Laos ethics protocols. Data collection with children (aged 0–17 years, or 14–17 years and unmarried) will require both child assent and caregiver consent.

Ethical approval was granted for both in-person and remote data collection and consent taking. In-person data collection is the main and preferred strategy. However, remote data collection is needed in some instances as severe weather events and poor roads limited access by the research team to participants. So far, response rates for remote data collection have been adequate. For in-person data collection, the information sheet will be either read aloud or given to the participant to read and written consent will be sought (signature or thumbprint if illiterate). For remote data collection (eg, by phone/video conferencing), an information sheet will be read aloud to participants. Oral consent will then be sought, and recorded (on a separate file to any interviews). Children will be read a simplified information sheet and will provide written or oral recorded assent (for in-person and remote interviews, respectively). The informed consent and assent processes will make it clear that the decision to participate, to withdraw or to refuse to answer any questions will not have any negative consequences, including on any pre-existing or future services they are receiving from UNICEF, the Laos government or any other provider. Furthermore, participants' responses and their decision to participate or not will not be shared with UNICEF, the government or anyone outside of the research team.

Collected data will be anonymised using a unique participant ID. A link log which links the unique ID to name will be kept on a secure server in a password-protected file, separate from the data. All study staff working with the data have undergone ethics training and will sign a confidentiality agreement. The study will comply with the General Data Protection Regulation, which requires that personal data must not be kept as identifiable data for longer than necessary for the purposes concerned. We will destroy all personally identifiable data 2 years after the end of the study.

Preliminary findings will be shared with UNICEF Laos, the Laos government and study participants for feedback. Findings from the trial will be published in academic journals and in short briefs targeted to a non-academic audience. Online and in-person dissemination events (eg, conferences, webinars) will be conducted in Laos, the UK and elsewhere.

### DISCUSSION

The UNICEF Laos/Laos government cash-plus programme for children with disabilities is an important programme to evaluate. Importantly, little information exists on the impact of social protection, particularly cash-plus programmes, among people with disabilities.[38] Information is particularly lacking about such

programmes among children, as studies are often underpowered to disaggregate by age.[17] However, access of children with disabilities and their families to social protection is particularly critical, as it may promote greater access to health, education and other support services that can have knock-on effects throughout their life-course. Information about the impact of this new, pilot programme, combined with a process evaluation detailing areas for strengthening, can help to inform improved delivery of this and other programmes in the future.

Still, there are some limitations of this evaluation. First, it was not feasible to conduct a non-randomised controlled trial. We will check for balance between the intervention and control arms and adjust analyses if there are differences between the two groups. However, it is possible that there are unobservable differences between arms that have not been accounted for. Second, some impacts of the programme may not be apparent after a 2-year follow-up period, particularly if there are delays in the roll-out of some components to the intervention. However, other evaluations of social protection programmes have found impacts on similar outcomes during this time frame.[17 39] Finally, some analyses, such as disaggregation by gender, may be underpowered depending on the final sample size.

Despite these possible limitations, this non-randomised controlled trial of a cash-plus programme will contribute to a limited evidence-base. Governments, international organisations and other actors have cited limited resources and evidence, as a barrier to investment in disability-targeted programmes. The lack of evidence on the effectiveness of social protection among people—particularly children—with disabilities hinders informed policymaking and planning, a gap which can be filled by the evaluation of a promising cash-plus programme.

**Author affiliations**
[1]London School of Hygiene and Tropical Medicine, London, UK
[2]Lao Tropical & Public Health Institute, Vientiane, Lao People's Democratic Republic
[3]London School of Hygiene and Tropical Medicine Faculty of Infectious and Tropical Diseases, London, UK
[4]Independent Consultant, Bhangaghar, India
[5]Department of Global Health & Development, London School of Hygiene and Tropical Medicine, London, UK
[6]Clinical Research, London School of Hygiene and Tropical Medicine, London, UK

**Acknowledgements** We are thankful to staff at UNICEF Laos, particularly Maryam Abdu and Amphayvan Chanmany for their contributions in informing the study design and identifying participants.

**Contributors** LMB, BS, NS, LS, MTC, TS, SC, CD, DG, AZ, HK, KC contributed to the design of this protocol. LMB wrote the original draft of this paper and BS, NS, LS, MTC, TS, SC, CD, DG, AZ, HK, KC critically reviewed and approved the paper. LMB, BS, KC, NS, MTC, LS contributed to the development and review of study tools. HK obtained funding.

**Funding** This research was funded by the United Kingdom's Foreign, Commonwealth and Development Office (PENDA project, grant PO8073). HK's salary is funded by an NIHR Global Research Professorship. LMB's time is partially covered by an Arts & Humanities networking grant (AH/X009580/1) and the Open Access fee is covered by this grant.

**Competing interests** LMB has previously worked as a consultant for UNICEF and holds grants in which UNICEF is a collaborator. None of this work is linked to this evaluation and is through UNICEF headquarters rather than UNICEF Laos.

**Patient and public involvement** Patients and/or the public were involved in the design, or conduct, or reporting, or dissemination plans of this research. Refer to the 'Methods' section for further details.

**Patient consent for publication** Not applicable.

**Provenance and peer review** Not commissioned; externally peer reviewed.

**Data availability statement** Data are available on reasonable request. We are investigating if it will be possible to anonymise the data in a way that would allow for reproduction of findings without identifying participants.

**ORCID iDs**
Lena Morgon Banks http://orcid.org/0000-0002-4585-1103
Nathaniel Scherer http://orcid.org/0000-0003-1846-8691
Hannah Kuper http://orcid.org/0000-0002-8952-0023

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
