## [Reviewer comments · BMJ Open]

ARTICLE DETAILS

TITLE (PROVISIONAL)	Impact evaluation of a cash-plus programme for children with disabilities in the Xiengkhouang Province in Lao PDR: Study protocol for a non-randomised control trial
AUTHORS	Banks, Lena Morgon; Soukxaphone, Bounhome; Scherer, Nathaniel; Siengsounthone, Latsamy; Carew, Mark T.; Shakespeare, Tom; Chen, Shanquan; Davey, Calum; Goyal, Divya; Zinke, Anja; Kuper, Hannah; Chanthakoumane, Ketmany

VERSION 1 – REVIEW

REVIEWER	Anderson, Kirsty Florida State University College of Social Work
REVIEW RETURNED	19-Dec-2023

GENERAL COMMENTS	This protocol addresses a very important aspect of quality of life for people with disabilities and their families (financial well-being) and focuses on a population that is understudied in extant literature (Laos). The protocol is well-written and clear, overall, and I have no major revisions. Minor suggestions are as follows: Introduction and Rationale:  - The authors did a nice job describing the gaps in extant literature, and the need to focus on outcomes beyond food insecurity, health status, and health seeking behavior. The authors also justified the need for cash assistance (as indicated by higher expenditures incurred among families/people with disabilities). However, it was unclear why the program also focused on the provision of rehabilitation and assistive devices in the introduction. A quick discussion describing what these devices are, and why they are essential to the quality of life of children with disabilities, would be useful. Particularly any empirical evidence indicating these devices are an “unmet need” would help to rationalize objective 3. - Likewise, the rationale for the project would be strengthened if the authors provide empirical support indicating that caregivers well-being is also impacted (in addition to income-related challenges, like poverty). For instance, we know that poverty and caregiver burden is associated with poor mental health among caregivers. Including this information could help contextualize the problem to readers unfamiliar with the disability population, and also rationalize objective 2. - The authors state, “evidence is needed on the impact of different programmes from different settings, and amongst children with disabilities in particular.” To provide context to an international audience, could the authors include information about the prevalence and distribution of disability in Laos? Are there specific national policies that might be useful for the reader to know? In other words, why is a context-specific assessment in Laos especially
--

	important and how might findings from this evaluation translate to other countries or populations? - I also recommend some clarifying text be added to the methods section:  o Intervention Description. For clarification, what are some examples of assistive technologies that you assessed for? The authors list specific domains on page 11 (healthcare, assistive devices, inclusive education supports, personal assistance) but does not define/operationalize them. o Eligibility criteria and recruitment. For clarification, could you elaborate how case ascertainment was validated? How did you determine the person had a disability? Does the study use the same definition of “disability” as the UNCRPD? How did you identify potential recruitments? What was your sampling frame? o Measures. The authors clearly describe their outcome measures and use of validated scales. But could benefit from a clearer description of process-oriented measures (acceptability of the programme, challenges and enablers to its delivery and mechanisms of impact).
--	---

REVIEWER	Pereira, Éverton Universidade de Brasília, Public Health
REVIEW RETURNED	24-Jan-2024

GENERAL COMMENTS	The protocol presented is interesting and presents relevant elements for reflection on disability and evaluation of social policies for this audience. Properly, it points out the need to produce evidence on evaluation and presents a protocol that can inspire other researchers and/or managers. However, I consider that the text still presents important gaps in information. I suggest a review to improve the quality of production. Specific comments are in the attached document – contact publisher to view.
--

VERSION 1 – AUTHOR RESPONSE

Reviewer 1

- 3) **However, it was unclear why the program also focused on the provision of rehabilitation and assistive devices in the introduction. A quick discussion describing what these devices are, and why they are essential to the quality of life of children with disabilities, would be useful. Particularly any empirical evidence indicating these devices are an “unmet need” would help to rationalize objective 3.**

We have now added more details to the introduction and have also provided a theory of change under intervention description.

- 4) **Likewise, the rationale for the project would be strengthened if the authors provide empirical support indicating that caregivers well-being is also impacted (in addition to income-related challenges, like poverty). For instance, we know that poverty and caregiver burden is associated with poor mental health among caregivers. Including this information could help contextualize the problem to readers unfamiliar with the disability population, and also rationalize objective 2.**

As with comment 3, we have now added this information to the introduction and the conceptual framework.

- 5) **The authors state, “evidence is needed on the impact of different programmes from different settings, and amongst children with disabilities in particular.” To provide context to an international audience, could the authors include information about the prevalence**

and distribution of disability in Laos? Are there specific national policies that might be useful for the reader to know? In other words, why is a context-specific assessment in Laos especially important and how might findings from this evaluation translate to other countries or populations?

We have now added a paragraph of background on Laos.

- 6) Intervention Description. For clarification, what are some examples of assistive technologies that you assessed for? The authors list specific domains on page 11 (healthcare, assistive devices, inclusive education supports, personal assistance) but does not define/operationalize them.**

We have now added some examples of assistive devices. We are unsure the full list of devices that UNICEF will provide, but this will be explored in greater detail during the process evaluation.

- 7) Eligibility criteria and recruitment. For clarification, could you elaborate how case ascertainment was validated? How did you determine the person had a disability? Does the study use the same definition of “disability” as the UNCRPD? How did you identify potential recruitments? What was your sampling frame?**

We have now added the following: “Disability determination and identification for both the intervention and control groups was conducted by UNICEF Laos. An initial sample – which strove to identify all children with disabilities in the study areas – involved a combination of past survey data and identification conducted by community leaders; these children were then verified by UNICEF to ensure they had a disability and were of the target age. Further details on the identification process will be explored as part of the process evaluation.”

- 8) Measures. The authors clearly describe their outcome measures and use of validated scales. But could benefit from a clearer description of process-oriented measures (acceptability of the programme, challenges and enablers to its delivery and mechanisms of impact).**

We have now added a section describing the process-related measures.

Reviewer 2

- 9) Could be nice write about the difficult to people with disabilities get jobs in many countries. It could explain as well how important is the social protection for childrens and families.**

We have added the following: “People with disabilities are often excluded from opportunities for decent work, due to discrimination or earlier barriers to receiving quality education and training, while caregivers often reduce time on income-generating activities for additional caretaking responsibilities.”

- 10) There are many different ways that countries implant social protection for people with disabilities. The autors could explain a little bit more about some countries and the difference between them.**

We have added the following line: “The coverage and content of these programmes can vary widely: for example, only 9% of people with severe disabilities receive a disability benefit in low-income countries compared to 86% of people with severe disabilities in high income countries.”

- 11) And how the studies talk about differents forms of social protection in different countries?**

We have added more information about the content of the studies.

- 12) Think about children is think about the parents as well....**

We have amended this statement to include “and their households”.

- 13) It is very imporant, but the authors don’t talk about how we need to change the society as well as give “cash transfer” only for individuals. I think could be nice if them write about those theories of disability.**

We have added the line: “Further, discrimination and negative attitudes on disability can limit participation and well-being”.

- 14) The authors need to talk more about Laos: what the country is like, what types of protection are available for people with disabilities, what life is like for people with**

disabilities, etc. I consider this information essential for a better understanding of the reasons that led the authors to adopt the form of evaluation presented in the protocol.
We have now added a paragraph about the context in Laos.

15) How is the healthcare system in Laos? How do people access it? How the system organization influence the need for extra income for people with disabilities? There is a lack of information about Laos, as I said earlier.

We have now added information about the Laos health system.

16) Wellbeing can be considered a relational and cultural/social concept. It is important in an instrument to think about possible influences from the history, economy, culture and form of organization of the place in which it will be used. I suggest a deeper debate about this, problematizing possible biases of the instrument considering the context of use.

We have added the following: "It included commonly used measures, such as ones used in the Multiple Indicator Cluster Surveys that have been administered in over 100 countries including Laos."

17) Depending on the place of use, interpretations and/or local forms of life can give different meaning to what you want to investigate. I suggest detailing where this instrument was used before and what are the similarities between the places of use and Laos.

In adding more detail on the process evaluation, we have explained that we will triangulate the quantitative data with qualitative in order to have a more nuanced understanding.

18) Explain the reasons for using Vietnam as a parameter

We have clarified: "Disability-sensitive MODAs are lacking, but using data from a previous study in Vietnam as parameters..."

19) Will they be residents of Laos? Tell us more about the people who will collect the data.

We have clarified: "They are predominantly public health students at the Lao TPHI and speak Lao and/or Hmong."

20) There is no detail on what the in-depth interviews will be like, how they will be carried out, who will carry them out, etc. Furthermore, it is necessary to think about how they will compose the proposed evaluation. We also do not have information on how in-depth interviews were analyzed. The authors do not discuss the qualitative stage of the protocol. The description of the method privileged the quantitative aspects, not explaining how the qualitative stage will be carried out.

We have now added significantly more information about the qualitative process evaluation throughout.

21) The issue of collecting data remotely is also a little confusing. The authors need to detail that in the method, especially when, why, and how the online collections will be carried out. It is also necessary to problematize the scope of this form of collection considering the socioeconomic conditions of the country and the group studied.

We have added the following: "Ethical approval was granted for both in-person and remote data collection and consent taking. In-person data collection is the main and preferred strategy. However, remote data collection is needed in some instances as severe weather events and poor roads limited access by the research team to participants. So far, response rates for remote data collection have been adequate."

22) It is important to problematize the link between the research and the government of Laos. People can be afraid of the answers and this can mean biases in the data collected.

We have added the following: "Further, participants' responses and their decision to participate or not will not be shared with UNICEF, the government or anyone outside of the research team."

23) Nothing was said about how the qualitative stage (or in-depth interviews) will form the analysis framework. I imagine that this will form the analysis framework and also presents potential and limitations in the evaluation of the program. I suggest adding to the discussion.

We have now added significantly more information about the qualitative process evaluation throughout.